# Eliciting initial programme theories for a health research capacity strengthening initiative targeting African universities: A realist synthesis

**Meshack Nzesei Mutua**[1*], **Andrew Harding**[2], **Ferdinand C. Mukumbang**[3], **Justin Pulford**[1]

**1** Centre for Capacity Research, Liverpool School of Tropical Medicine, Liverpool, United Kingdom, **2** Division of Health Research, Lancaster University, Lancaster, United Kingdom, **3** Department of Global Health, University of Washington, Seattle, Washington, United States of America

* Meshack.Mutua@lstmed.ac.uk

## Abstract

There is a growing call for theory-driven evaluation approaches to health research capacity strengthening (HRCS) interventions. Specifically, realist evaluation has gained wide attention given its response to the question: How does an intervention work, why, for whom and under what conditions? In realist evaluation, initial programme theories (IPTs) are first elicited before they are tested and refined. This article describes the IPTs of an HRCS programme aimed at strengthening the research capacity of African universities. Using the 'Developing Excellence in Leadership, Training and Science (DELTAS Africa) programme as a real-world case, the IPTs were drafted through a review of the programme documents and other published literature. Seven programme documents and 32 published papers covering 26 research capacity strengthening initiatives in African universities were reviewed. Different Context, Mechanism and Outcomes (CMO) were extracted and CMO configurations were formulated. Thereafter, the CMO configurations were refined through four interviews with the DELTAS programme designing team. Three transferrable IPTs were elicited. Evidence suggests that, for HRCS interventions to be more effective in strengthening research capacity in African universities, systemic challenges (e.g., lack of funding for health research, ineffective research policy environment and lack of institutional support for research) need to be addressed and the university staff/ researchers empowered, incentivised and motivated. The article underscores the importance of institutional buy-in, effective implementation of research policies (e.g., protected time, research career pathways, gender equality, research ethics and integrity, anti-bullying and anti-harassment, etc.), long-term research funding and equitable research partnerships in fostering a strong research environment and culture. Notably, the article makes a methodological contribution by demonstrating how IPTs can be developed using disparate evidence sources. The IPTs will be tested and refined through a








**Data availability statement:** All relevant data are within the manuscript and its Supporting information files.

**Funding:** This work has been funded by the LSTM-Lancaster Medical Research Council (MRC) Translational and Quantitative Skills Doctoral Training Partnership (DTP) Award. The funders had no role in study design, data collection and analysis, decision to publish, or preparation of the manuscript.

**Competing interests:** The authors have declared that no competing interests exist.

primary realist evaluation, which will further refine the CMOs presented in this article and provide insights into the current HRCS evaluation framework.

## Introduction

Health research capacity strengthening (HRCS) has been defined by Lansang and Dennis [1] as the "ongoing process of empowering individuals, institutions, organisations and nations to: define and prioritise problems systematically; develop and scientifically evaluate appropriate solutions and share and apply the knowledge generated" (p. 764). HRCS is a global strategy aimed at improving the ability of researchers, research institutions and research users in low-and middle-income countries (LMICs) to carry out and promote the use of research evidence that is geared towards addressing health challenges and the needs of the population [2].

HRCS efforts have been categorised into three levels, namely, individual, institutional and environmental/ societal levels. Evidence shows that the majority of HRCS interventions happen at the individual level [3–5], including activities such as scholarships, specialised training (e.g., on research design, grant writing and publication), mentorship and industrial placements. These individual-level HRCS interventions are deemed not only as less complex [6] but also generate less impact compared to institutional and societal interventions [7]. Institutional-level HRCS interventions seek to strengthen the capacity of research departments or units in universities or research institutions to effectively play their role within the national health research system [8,9]. Examples include upgrading research infrastructure (e.g., laboratory equipment, computers and data management infrastructure), strengthening administrative capacity (e.g., in finance, grants and research management functions), and establishing career pathways for researchers [7]. Societal-level HRCS interventions seek to improve the research system by ensuring relevant research policies are adopted that will incentivise individuals and organisations to conduct high-quality research, stimulate positive political will and promote an effective regulatory environment for research [5]. Examples include efforts to improve the ability of research entities to manage transparent, efficient, and competitive processes for allocating national research funds and demonstrate research productivity (such as funds, publications, patents), expansion of networks/ collaborations by national/international research bodies, and policy engagements that influence research uptake [5]. The effects of institutional HRCS interventions interlink both individual and societal level capacities with ripple effects that extend beyond the research system [5,10], thus making such interventions complex and multifaceted in both design and implementation [11–15].

The biggest challenge, therefore, is to enhance the research capacity at institutional and societal levels sustainably, which potentially generates a greater impact at the systems level [16,17]. This is exacerbated by the fact that there is an underdeveloped evidence base on how to best design, implement and evaluate such interventions. The nature of the evaluation approaches, coupled with relatively few evaluations in HRCS, could partly be blamed for the existing evidence gap. Most

evaluations of the HRCS interventions have used traditional approaches, which do not provide evidence on how HRCS interventions work, across different contexts, to generate research capacity outcomes [18,19]. Yet, as complex, dynamic and multifaceted initiatives, HRCS interventions demand more complexity-aware evaluation approaches [6]. An appropriate evaluation for complex interventions such as HRCS should not only focus on what works but also provide answers to why, how, for whom, and under what conditions; this is grounded in a realist approach to evaluation.

Realist evaluation first emerged from Pawson and Tilley [20] seminal work grounded in scientific realism. The approach is suitable for uncovering rich and transferable insights into the effects of complex interventions, given its focus on generative causal mechanisms, which are context-dependent [20]. The essence of realist evaluation, according to Pawson and Tilley [21], is not to provide a verdict as to whether a programme has succeeded or failed, but rather to examine how a programme interacts with specific local contexts to trigger mechanisms necessary for generating intended or unintended outcomes. Essentially, realist evaluation uses the Context (C), Mechanisms (M) and Outcome (O) configuration as a tool to explore generative causation [22]. In practice, realist evaluation starts with a programme theory and ends with a refined programme theory [20]. Initial programme theories (IPTs) should be elicited first before they are tested and refined.

## Methods

In this article, we describe the IPTs of an HRCS programme aimed at strengthening institutional health research capacity, specifically in African universities. The aim is to elicit transferrable IPTs that can be applied in African university settings to inform the design, implementation, and evaluation of HRCS interventions, thus ensuring that they can drive sustainable impact. A detailed protocol for this synthesis has been published elsewhere [23]. The IPTs were developed through a

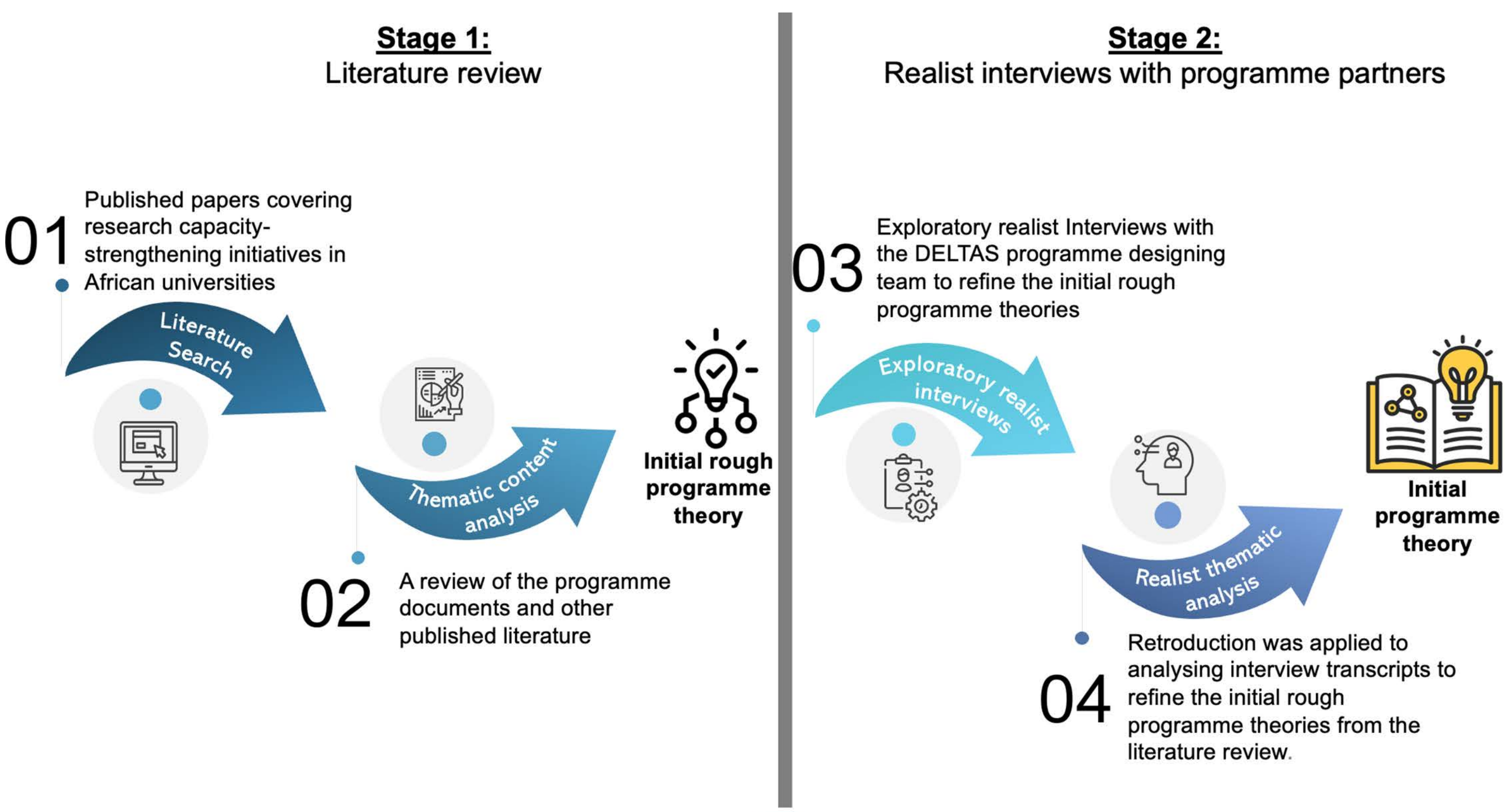


**Fig 1. Theory gleaning process.**


phased realist synthesis process involving a) a realist review of HRCS programme documents and published literature on research capacity strengthening of African universities, and b) interviews with the programme's designing team/partners – see Fig 1 below. The 'Developing Excellence in Leadership, Training and Science (DELTAS Africa) programme, one of Africa's HRCS flagship initiatives, was used as a real-world HRCS case.

### The DELTAS Africa programme

The DELTAS Africa programme is a science and innovation initiative funded ($100 million) by the Wellcome and the Foreign Commonwealth and Development Office. The first phase of the programme (2015–2021) supported eleven [11] research groups/ consortia to carry out collaborative research, train young and emerging researchers to be scientific leaders and strengthen the research capacities of universities and research institutions in Africa. By doing so, the programme aims to deliver cutting-edge and impactful health research in the long term. The first phase programme was jointly implemented by the Alliance for Accelerating Excellence in Science in Africa (AESA) based at the African Academy of Sciences (AAS) and the African Union's New Partnership for Africa's Development (NEPAD). Although phase two of the programme runs from 2023 to 2026, this study will focus on the first phase since research capacity strengthening outcomes can take a long time to be realised. Importantly, the study will not be evaluating the DELTAS Africa programme as a whole but rather the institutional research capacity-strengthening component of the programme.

In terms of structure, the paper first describes the realist review process, the review findings and how the draft IPTs were drafted by framing a causal link between context, mechanism and outcomes derived from the reviewed papers. Findings from the interviews (with programme designers) are then presented, and a description of how the IPTs were further refined is provided. The CMOs are then discussed – bringing in additional perspectives from published literature to shed more light on institutional HRCS practice – and a conclusion is presented.

### Stage 1: Realist review

Realist reviews/syntheses have been used to both develop IPTs [24] and test and refine them [25–27]. In this study, the review process aimed at identifying contexts, mechanisms and outcomes related to institutional research capacity strengthening (in African universities) that have been described by a) published literature and b) the DELTAS Africa programme documents. The review sought to address three questions:

1. What are the key mechanisms that drive the outcomes of health research capacity strengthening for universities in Africa?
2. What are the contextual factors that enable the triggering of mechanisms necessary for generating health research capacity outcomes for African universities?
3. What is the causal connection between the key mechanisms, contexts, and health research capacity outcomes?

**Search strategy and selection of papers.** Firstly, relevant DELTAS Africa programme documents were requested from the DELTAS programme staff, and seven [7] documents, including grantee funding proposals, programme Theory of Change (ToC) and M&E Framework documents, were shared. A few potentially relevant documents, e.g., grantees' progress reports and consolidated programme reports, were not shared as the programme staff cited a lack of copies of the reports. The programme documents were reviewed to gather insights about how the programme was designed and to identify the underlying assumptions. The documents mainly described the programme architecture and partially described the desired research capacity outcomes and some contextual elements. The ToC document presented an overarching high-level model of DELTAS programmatic domains, namely scientific quality, research leadership, scientific citizenship and research management, culture and infrastructure and their related activities, outputs, and outcomes. Notably, none of the documents articulated or described mechanisms, though this was unsurprising because the programme was not

designed with a realist lens. However, through the review process, a better understanding of the DELTAS Africa activities, who was targeted by them, and what effect or outcomes were generated was achieved. This information was extracted as themes and analysed and synthesised together with the information extracted from the published literature (see below).

Secondly, the published literature was identified through searches on electronic databases. Specifically, this involved a search on Google Scholar, Global Health, PubMed, and Web of Science using the search terms (((“research capacity strengthen*” AND “university*” AND “Africa”) AND (“research capacity build*” AND “university” AND “Africa”) AND (“research capacity develop*” AND “university” AND “Africa”))). The *Health Research Policy and Systems* journal, which publishes scholarly work focused on health research capacity, systems, and policies, was also searched. The journal was searched using two keywords, i.e., “university” and “Africa”, and then the results were screened together with the rest of the records identified from databases. The search included literature published in English and from the earliest records available in the databases up to 10 February 2024. EndNote 20 was used to organise and screen the results/references.

Papers were considered eligible for inclusion if they reported 1) research capacity strengthening programmes/ initiatives, 2) the initiatives were implemented any time prior to 2024, 3) in an African setting, and 4) a university was (one of) the study setting. Different publication types, such as empirical studies and commentaries, were included.

**Relevance and quality appraisal.** Realist reviews significantly differ from the conventional Cochrane reviews such as narrative reviews and meta-analyses [24,28]. Dada and colleagues argue that the “inclusion criteria and appraisal of evidence within realist reviews depend less on the methodological quality of the study and more on its contribution to our understanding of generative causation, which is uncovered through the process of reproductive theorising” [29]. Since realist review quality appraisal does not hierarchise evidence [24,28], any study types and available evidence can be included to glean the initial programme theory [24]. Instead of relying on tools such as the Mixed-Methods Appraisal Tool (MMAT) and the Critical Appraisal Skills Programme (CASP), realist reviews emphasise the three Rs: rigour, relevance, and richness [29].

The realist review appraisal form by Molitor et al. [24] was used to assess the relevance and quality of the selected papers and how they contribute to the theory-gleaning process (see S1 File). Relevance, richness and rigour of the evidence are key principles in realist appraisals [24,28,29]. The relevance principle asks whether the paper (or sections of it) corresponds to the review question(s); the rigour principle asks if the quality of the data is good enough to be included, and richness is concerned with the contribution of each paper to theorising [29,30]. The appraisal process was guided by two key questions: a) does the paper focus on research capacity strengthening within an African university, and b) does it present/ describe useful information about context, mechanism, and outcomes? As shown in the Supplementary Material (S1 File), all the included papers met the relevance and quality criteria and the process adhered to the methodological guidance and principles described by Wong et al. [31]. Although none of the included papers made the connection between the context, mechanism, and outcomes, each paper described at least one of the CMO components. The majority of the empirical papers included in the review provided a detailed description of the research objectives, methods, and participants to the extent that the study could be replicated. Notwithstanding, the insights and themes extracted from the commentary papers were triangulated with those extracted from the empirical papers, thus strengthening the review output.

**Data extraction and analysis.** An Excel template was used to extract data on the characteristics of the included papers and the RCS programmes reported, including the first author’s affiliation, the subject focus, the RCS levels (individual, institutional or societal) focused on by the paper, the funding source and the research methods employed by the studies. The realist review appraisal form developed by Molitor et al. [24] was used to extract data on the context, mechanism, and outcomes of RCS initiatives, which allowed the articulation of the connection between the outcomes and process [C + M = O] and the articulation of rival theory. Each paper was read twice to become fully familiar with the themes.

Thematic analysis was applied, which allowed for the systematic identification of recurrent or salient themes across the included papers [32]. The themes were then categorised as mechanism, outcomes, and context. Since the extraction of

the themes was not guided by already defined programme theories, inductive reasoning was applied, and this helped to ensure that alternative ways of making sense of the data were not overlooked [33,34]. Once the extraction of the themes was completed, the data were organised in a three-column table with Column 1 capturing 'Context', Column 2 'Mechanism' and Column 3 'Outcomes'. While this cataloguing approach is generally discouraged in realist-informed studies [35], we adopted it because it allowed us to start the process of gleaning an initial draft programme theory. This reorganisation, combined with retroductive reasoning, facilitated the analysis and synthesis of the data [36]. A synthesis of the extracted data was carried out, identifying commonalities, differences and associations across themes and papers, and this informed the framing of the CMO configurations. Discussions between the co-authors, whose expertise interfaces the realist approach and HRCS, helped with sense-checking throughout the process.

**Description of included papers.** The electronic search resulted in 1439 papers – see Fig 2. After removing duplicates, 414 records (see S2 File) were screened against the eligibility criteria. The full texts for 88 records were retrieved and read, out of which 32 papers (19 empirical and 13 commentaries) were included for review (see S3 File). S4 File provides details of how the review adheres to the guidelines for reporting systematic reviews.

**Summary characteristics of the included papers.** The papers were published between 2010 and 2023, describing RCS programmes or initiatives that were implemented between the years 1998 and 2022. The 19 empirical papers included 13 papers that employed qualitative methods (in-depth interviews), 3 papers that employed mixed methods (surveys and in-depth interviews), 2 papers that involved review or analysis of secondary evaluation data and 1 paper that employed quantitative method (survey) only. The 13 commentaries included authors' reflections on the experience

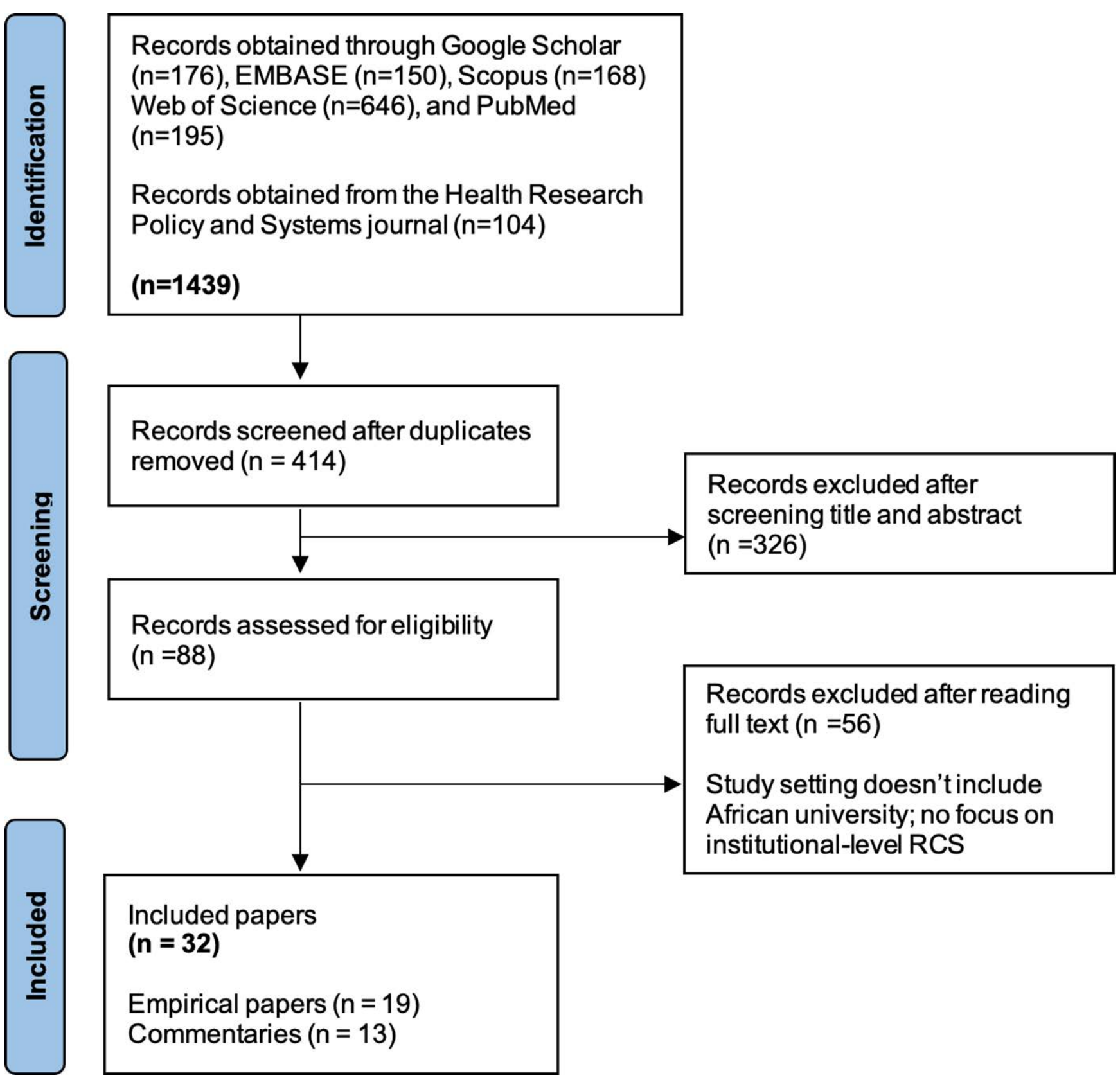


**Fig 2. Papers selection and screening process – adapted from Page, McKenzie [37].**


of implementing an RCS initiative, the successes, challenges, and enabling and hindering factors. As summarised in Table 1 below, 21 papers were first authored by researchers affiliated with an African research institution/university, while 11 papers were first authored by individuals affiliated with non-African institutions. 28 papers were published in journals based in the Global North, and 4 were published in African-based journals.

As shown in Table 1, 26 papers described programmes that were funded by Global North partners, while in 6 papers, the source of funding was either not reported or anonymised. Different initiatives were led by different partners, with 16 of the programmes involving South-North partnership, 8 initiatives involving North-South partnership and 5 initiatives involving South-South partnership. 2 papers focused on RCS activities at the institutional level only, 21 papers focused on RCS at both individual and institutional levels and 9 papers focused on RCS at three (individual, institutional and societal) levels. 26 papers focused on RCS in health disciplines or disease-specific research areas (e.g., HIV/AIDS, malaria, or mental health), and 6 papers focused on RCS in general. Notably, it was impossible to determine the number of universities covered across the 32 papers and where they were based (regionally) because 12 papers had anonymised the names of the universities and the African region in which they are located.

### Stage 2: Interviews with programme partners

Four [4] interviews with the DELTAS Africa Phase I programme designing team representatives were conducted to further refine the IPTs. The interviews were conducted between June and July 2024. The aim was to elicit their understanding of how the DELTAS programme (one of the flagship multi-million dollar and long-term RCS initiatives) was envisioned to work to generate the desired research capacity outcomes and how context moderates or mediates the outcomes. The participants possessed extensive experience implementing similar RCS initiatives in Africa, and therefore, the insights they provided were not limited to the DELTAS programme only. The realist interviewing technique was applied, thus focusing the discussions on the IPTs [38].

After the interviews were conducted, the interview recordings were transcribed verbatim. The transcripts were then keenly read (first-level analysis) to gain a holistic understanding of the data. Afterwards, the transcripts were re-read

**Table 1. Characteristics of the included papers.**

| Category | Characteristic | Description | Frequency and Percentage |
|---|---|---|---|
| Publication characteristics | Type of publication | Empirical | 19 (59.4%) |
| | | Commentary | 13 (40.6%) |
| | First author affiliation | African | 21 (66.6%) |
| | | Non-African | 11 (34.4%) |
| | Journal | African based journal | 4 (12.5%) |
| | | Non- African based journal | 28 (87.5%) |
| Programme characteristics | Funding source | Global North funding partner | 26 (81.3%) |
| | | Not reported or anonymised | 6 (18.7%) |
| | Subject focus | Health and disease-specific research capacity | 26 (87.3%) |
| | | General research capacity | 6 (18.7%) |
| | Primary RCS levels | Institutional only | 2 (6.3%) |
| | | Individual and Institutional | 21 (66.6%) |
| | | Individual, Institutional and Societal | 9 (28.1%) |
| | Nature of programme partnership | South North | 16 (50.0%) |
| | | North-South | 8 (25.0%) |
| | | South-South | 5 (15.6%) |
| | | Unreported | 3 (9.4%) |


(second-level analysis), and fragments of data were accordingly coded as context (C), intervention (I), mechanism (M) or outcomes (O). Retroductive theorising was applied to data analysis, employing both deductive and inductive perspectives. Deductive reasoning was applied to identify context, strategy, mechanism and outcomes (CMOs) that were aligned with the IPTs, and inductive reasoning was applied to identify potentially new CMO elements (not captured by the IPTs). The ethics clearance for this study was granted by both the Liverpool School of Tropical Medicine, UK, and the Strathmore University, Kenya, Research Ethics Committees as part of a wider realist study. Informed consent (verbal) was obtained from the participants.

### Findings from the literature review

The sub-sections below present the CMO elements extracted from both the published literature and the DELTAS Africa programme documents. Where necessary, excerpts of the programme documents were extracted and included in the review findings. Since the aim was to extract potential CMO elements from the literature for the theory-gleaning purpose, both actual and aspirational contexts, mechanism and outcomes were extracted. In this article, the actual CMO elements were those that were measured by the RCS initiative, and the aspirational elements were those desired/ expected for the programme to be effective (or not).

## Mechanism

Although the focus remains 'research capacity strengthening at the institutional level', the reported mechanisms were all triggered at the individual level. Multiple papers described the mechanisms of *trust* and *respect*, *commitment* and *enthusiasm*, *motivation* and *incentives, empowerment of faculty staff*, and *ownership* and *buy-in* by institutional leaders that were triggered in specific contexts to generate RCS outcomes.

### Trust and respect

Three papers (9.4%) reported trust and respect between researchers or research partners as an important mechanism that enables collaborations and partnerships in RCS to thrive, [39–41] whether those collaborations or partnerships are between researchers from the Global North and African universities or between African universities. One of the papers highlights that partnerships/ collaborations that are geared towards strengthening research capacity must appreciate and respect the local knowledge and priorities [40].

### Commitment and enthusiasm

Three papers (9.4%) highlighted commitment and enthusiasm as a central mechanism for university faculty and post-graduate trainees taking on research careers and leveraging research collaborations [42–44]. The papers highlighted that researchers and research leaders must sustain the commitment and enthusiasm if a collaboration or partnership is to remain stable and deliver positive RCS outcomes [43,44].

### Motivation and incentives

Seven papers (21.9%) reported that the RCS programmes would be effective if the faculty staff are highly motivated, willing, and incentivised to drive substantive and sustainable institutional change at African universities [45–51]. The papers highlighted that this would work in contexts where the universities provide research grants to faculty staff as incentives to retain them [46] and where partnerships allow the sharing of mutually beneficial research resources (e.g., sharing of lab space, research space and equipment), which is crucial for learning and research production [49]. However, one of the articles indicated that the individuals have to be motivated and willing to transfer the research training and skills to their institutions [50], which is key in institutional capacity strengthening.

In the DELTAS Africa programme, the selection of researchers for fellowship opportunities (across the consortia) considers their motivation levels and career ambitions, and then the programme provides them with the resources and incentives to strengthen their research leadership and capacity. One of the documents captured the following:

> "Senior research fellows/ scientists who are motivated and have demonstrated high research productivity and leadership potential will be supported to consolidate their independent research and build a group at African partner universities" [DELTAS Programme Document 4].

### Empowerment

Five papers (15.6%) highlighted that the empowerment of researchers – which enhanced their sense of agency – was key to translating their individual technical and soft skills and competencies to institutional capacity [47,52–55]. The respective RCS programmes aimed to impart – through training – committed and motivated researchers with wide ranging technical and soft skills, but the individual must believe in their personal capabilities and exercise their agency if high quality research is to be produced and institutional research goals met [55]. By exercising their sense of agency, individually or collectively, the university staff and researchers will positively use their competencies and skills both to their advantage and the institution's [53].

### Ownership and buy-in by institutional leaders

Twelve papers (37.5%) have reported ownership and buy-in by institutional leaders as an important mechanism that influences the effectiveness of institutional RCS [39,41–43,45,47,49,53,55–58]. This mechanism may be triggered in the context where equitable and strategic partnerships are developed and where institutional leaders feel that the RCS initiatives are addressing local priorities and needs and, importantly, have the autonomy or influence on how the programme resources are utilised [39,41,45,47].

As the institutional research capacity improves and the RCS initiatives improve institutional practices and culture, this mechanism may become part of the context where ownership and buy-in from university management and faculty staff would be a prerequisite for RCS initiatives to thrive [39,46,56,59,60]. In such a case, for instance, RCS efforts that involve introduction and implementation of new policies, systems or initiatives will require institutional buy-in and support from the leaders if such efforts are to be institutionalised and sustained [56]. Johnson and Louw [53] posit that, institutional leaders such as deans, heads of departments and senior academics need to be engaging, supporting, and encouraging faculty to pursue high quality and impactful research if the research culture is to grow (p. 159). Leadership practices were central to the provision of strategic direction which facilitated institutional-level RCS [56]. Strong leadership allowed ownership that facilitated the introduction of necessary change in local institutions [57].

The DELTAS programme documents highlighted the critical role that institutional leadership play in RCS efforts and key strategic decisions such as allocation of resources. Ensuring that these leaders are on board is crucial if RCS initiatives are to be effective. One of the programme documents captured the following:

> "Vice Chancellors (VCs) are in a better position to support the institutionalisation plans that [researchers/staff] may be pursuing within their institutions. Examples of this include institutional investment in research infrastructure, changes to PhD training programmes at institution level, changes to promotion criteria, attention to academic integrity and taking on harm reduction strategies. … Regular communication/ engagement of the institutional leadership (e.g., through the biennial meetings with the VCs to present evidence on best practice learned and challenges encountered) will help garner support from the institutional leadership and achieve a buy-in" [DELTAS Programme Document 2].

## Outcomes

The identified outcomes include increased research funding secured, improvement of research infrastructure and equipment, strengthened research management and governance systems, strengthened research partnerships, increased retention of research staff, research career and portfolio growth and improved health outcomes and policymaking. Notably, 'strained teaching and administrative resources' was identified as an unintended negative outcome. Each of these is described below.

### Increased research funding

Eleven (34.4%) papers reported securing additional funding as a result of improved research capacity at both individual and institutional levels. As the grant writing skills and research leadership of the university faculty/ researchers improved, they engaged in the development of multidisciplinary research grant writing, which resulted in additional funding being secured [40,42,44–46,48,49,52,54,56,58]. Notably, as the researchers continued to expand their networks, identify new collaborators, and develop collaborative grant applications, they secured collaborative research grants [44,45,49] which contributed to the institutional overheads.

### Improvement of research infrastructure and equipment

Eight papers (25%) reported 'improvement of physical research infrastructure and equipment' as an RCS outcome. Improvement of research infrastructure included procurement and accreditation of laboratory equipment [10,39,40,45,56], upgrading general infrastructure such as research office spaces [60,61], and installing interactive communication technologies that enabled the scaling up of training and research in the university [57]. The long-term impact of the infrastructure grant is that the improved research infrastructure and equipment remain at the universities for use by faculty long after the grant has ended [45].

The DELTAS Africa programme provides for infrastructure grants, which contribute to the upgrading of the research infrastructure and procurement of research equipment. One of the programme documents captured the following:

> At the institutional level, partner universities may apply for "infrastructure grants that have in the past been used to upgrade research training infrastructure like postgraduate seminar rooms and libraries. The support has also been used to upgrade internet connectivity and ICT support equipment (projectors, computers, printer/scanner), and renovating training facilities, and purchase research equipment, e.g., desktops/laptops in selected universities and anti-plagiarism software and statistical analysis packages" [DELTAS Programme Document 2].

### Research management and governance systems strengthened

Thirteen papers (40.6%) reported that the RCS programmes had resulted in strengthened research management and governance systems of the participating institutions. This included development and adoption of research and grants management policies and procedures [39,59], establishment of new research support units [41,42,57,62] and improvement of research ethics processes [16,40,44,49,53,54,57,60]. Collaboration/partnerships that involve establishment of research support centres/units deliver a long term impact [53] as those units can support the wider research community beyond the originally intended university departments. With strong research management and governance systems in place, other ripple effects can be realised such as production of high-quality and impactful research and research outputs which can ultimately improve the institutional research profile [53].

> "Programme provides funds to establish 2-3 research hubs. These funds may be utilized to equip a research management office (computers, printers, router), improve seminar or meeting rooms to support research and research training activities, or provide seed funds for a research hub to buy small and medium-sized equipment. … In terms of software,

the programme provided the initial institutional subscription to the anti-plagiarism software Turnitin with the partner institutions making a commitment to continue the subscription. The use of this software has now been incorporated in postgraduate training in all our partner institutions and so no further investments are necessary" [DELTAS Programme Document 2]

Research Management/Support Offices/Units within the participating universities will drive the adoption of existing good practices and support their implementation at the institutional level in the longer term. [DELTAS Programme Document 3]

Research institutes may have more established research management and governance infrastructures than universities in Africa [43]. Collaboration between the two can help the university adapt and institutionalise research management functions (establishment of research support units), which can help with recovering costs from future successful grants, thus making the university's research enterprise sustainable in the long term.

### Research partnerships strengthened

Five papers (12.5%) reported that institutional collaborations had been expanded and strengthened as a result of RCS efforts [44,45,56,60,62]. These were synergistic research collaborations and partnerships mainly between African and Global North universities and research institutions. Some of the collaborations/ partnerships became long-standing or institutionalised at the university and expanded to benefit other departments or faculty that were not originally targeted by the initiative [45].

### Strained teaching and administrative resources

One paper (3.1%) reported that some departments within the university had their teaching and administration resources overstretched because junior faculty staff had embarked on postdoctoral research training as part of institutional RCS [55]. As an unintended negative outcome, this meant that other faculty/staff had to take on additional responsibilities to cover for their colleagues who had transitioned to postdoctoral training. Though not an outcome of RCS, this theme was identified as critical since it is a consequence of junior university staff transitioning or growing in their research careers, which strains the available resources. Notably, this informs the framing of a rival theory.

### Increased retention of research staff

One of the papers highlighted that in a context where research career pathways are established, there is notable retention of faculty staff [62]. Faculty staff who are promoted into research leadership positions (in context where research career pathways are established) inherently become agents of change in their universities and support the transformations of their universities into research centres of excellence [51]. Established research career pathways in universities have also seen junior faculty staff undertaking PhDs, and postdoc fellows taking up research roles or being promoted to research leadership roles in the universities. The ripple effect of the established career pathways was the individual researchers contributing to publication, mentoring junior faculty, increased supervisory capacity, increased university faculty staff with PhD qualifications, and securing of research grants (which contributes to institutional overheads), and this illustrates the interlinkage between individual level capacity and institutional capacity [10,16,40,46,48,51,53,56,60,62].

In one of the DELTAS documents, it was reported that the RCS programme had introduced a new position into the university structure (postdoctoral research role) that was initially supported by the programme. The postdoctoral fellows were subsequently absorbed into the formal university structure. Below is the excerpt.

"The Consortium A team reported that discussions with University senior management about the need to regularise the postdoctoral position within the institutional faculty structure had positive result. Consequently, all the postdoctoral fellows from Phase I of the programme were absorbed into the University faculty system after completion of their

fellowship, which is an indicator of progress towards institutionalization of the postdoctoral fellow position" [DELTAS Programme Document 1]

#### Research career and research portfolio growth

Five papers (15.6%) reported that research fellows who had been trained and mentored through the RCS programme had progressed to occupy faculty positions and research leadership positions at the universities [40,42,50,52,61]. This outcome relates to previously described outcomes such as staff retention and securing of additional funding, which are consequences of individual level capacity being translated to institutional capacity. Notably, buy-in by institutional leaders and having established research policies, systems and culture in place was highlighted as a key contextual element that allows for individual capacity to be institutionalised [45].

#### Improved health outcomes and policymaking

Eleven papers (34.4%) have acknowledged 'improved health outcomes of the population' [16,41,42,47,55,63,64] and 'impacts on policy and practice' [16,41,52,61,65] as the ultimate and long-term outcomes of RCS. Although the authors recognise the limitation of their studies in that they did not document these long-term outcomes given that the studies were mostly conducted during the implementation of short-term RCS initiatives [45,49,56,66]; they highlighted that the goal of RCS is to generate impactful research that has utility in policy making and contributes to the improvement of people's health and wellbeing. To achieve these ultimate outcomes, Torondel et al. [47] posit that researchers and research entities should actively engage relevant actors throughout the research cycle and ensure research is translated and used by stakeholders.

### Context

The contextual elements identified include institutional research policies, systems, and culture; provisions for protected time for research; established research career pathways; established research infrastructure and equipment; reliable long-term funding; political and economic environment; and equitable and strategic research partnerships. Each of these is described below.

#### Institutional research policies, systems, and culture

Seventeen papers (53.1%) reported the centrality of institutional research policies, systems, and culture and how they enabled or hindered the effectiveness of RCS efforts and related outcomes [10,16,44,47,50,52–54,57–59,64,66,67]. Some of these research policies and systems may include provisions for protected time, establishment of research career pathways and research management and governance structures. Instruction-oriented universities are characterised by minimal financial and policy support for research and underdeveloped research governance and management policies [16,52,59], and navigating the complex university-level bureaucracy can be exhausting for researchers [16,67]. Since most African universities do not have established research policies and systems like other research institutions [16], interaction between universities and research institutes becomes an essential avenue for collaborative, impactful and sustainable research [64]. Established institutional research policies, systems and culture can catalyse, support, and incentivise the development of research [53], thus further strengthening the institutional research culture [50].

#### Provisions for protected time

Twelve papers (37.5%) reported the availability, or lack thereof, of protected time for research as a critical contextual element that determines the effectiveness of RCS activities. University faculty have to balance between their research

and teaching workloads, and mostly, their research careers and interests are stifled by intensive teaching workloads [16,39,42,49,50,53,55,57,64–67]. It is not a surprise that when junior faculty staff embark on doctoral or postdoctoral research training, it can result in unintended negative effects on the remaining faculty staff within their departments as they have to bear the additional teaching and administrative responsibilities relinquished by their colleagues [55].

For RCS initiatives to be effective, faculty staff must find a balance between research, teaching, and clinical and administrative duties [42], as this will allow them to spend more time in research and RCS activities. In a context where universities do not have provisions for protected time, RCS programmes are likely to be effective if they provide financial resources needed to buy out their teaching time, consequently enabling them to spend more time on research activities [39]. In fact, provision for protected research time by universities has been viewed as an indicator of how established the research culture is at universities [49]. One of the DELTAS Programme documents has highlighted that the participating universities are primarily teaching-centred, which means that faculty staff will spend more time teaching than in research work, and this explains why the research systems in those universities are less established. Below is a quote from one of the documents:

> "…universities where the institutional establishments are mainly teaching-centred and do not provide clear career paths for research-intensive personnel such as postdoctoral fellows. This serves as a disincentive for young scientists to take up such positions as an alternative to becoming lecturers and being burdened by high teaching loads which stifle their research careers" [DELTAS Document A]

### Established research career pathways

Six papers (18.8%) have highlighted how 'established research career pathways' in a university setting influence how well RCS programmes deliver institutional research capacity outcomes [44,48,55,65–67]. Established research career pathways are characterised by, for instance, the provision of tenure tracks to qualified research staff (alongside the academic career tracks on teaching and mentoring students) that help retain the talent [44,66,67] and provide professional development for research and research support staff [67]. Without the established research career pathways, some of the effects would include having few junior academic staff with PhD qualifications [48,55], a scarcity of senior faculty/researchers to mentor junior faculty staff, and a high burden on senior researchers to bear the institutional research workloads [65]. Established research career pathways ensure that there are adequate personnel contributing to a university's research pipeline. In some cases, this has been the primary strategy for strengthening research capacities in universities as the junior staff/ faculty are encouraged and motivated to pursue PhDs [63] and individuals are empowered to institutionalise research capacity in their universities [47]. In fact, Bates et al. [68] argue that paying attention to the skills of individuals is important even if the main focus of a programme is to strengthen institutional systems and processes (p. 7), while Aiyenigba et al. [56] – who examined how the DELTAS Africa Phase One was strengthening research capacities of the participating African institutions (including universities) – concluded that, besides direct infrastructural development, other institutional-level RCS benefits were driven through investment in individuals.

### Established research infrastructure and equipment

Eleven papers (34.4%) reported the availability of physical research infrastructure and equipment as central to institutional RCS [16,40,45,52,55,57,61,64–67]. Sewankambo et al. [55] have reported that the research spaces can improve social interaction among research staff, which can catalyse collaborative research activities. Dedicated learning spaces with strong internet connectivity are essential in promoting the research enterprise [52]. Importantly, since research institutions have more established research infrastructure and systems [16], universities that work closely with such research institutions will likely leverage their infrastructure [64]. The authors – across the 11 papers – consistently

argue that established research infrastructure and equipment allows high-quality research to be produced, and this has a ripple effect on the institutional research culture, and the established infrastructure is foundational for the acquisition of more funding resources. Modern communication technologies, for instance, that have been integrated into institutional infrastructure have supported open and fast sharing of information among researchers, thus facilitating collaborative research work [40].

### Reliable long-term funding available

Twelve papers (37.5%) reported that the RCS programmes were delivered in resource-constrained environments with limited or no funding from the local governments and primarily relying on external funding [16,40,42,44,45,48,50,51,56,61,66,67]. Limited resources affect not only almost all the RCS processes and the achievement of desired capacity outcomes but also the implementation of the research activities [42]. This lack of dedicated, long-term and sustainable local funding for research and RCS heavily undercuts both operational feasibility and human motivation [61]. It is unsurprising that authors have called for increased long-term financial investment towards RCS from governments, the private sector, and philanthropies if a significant impact on RCS should be realised [16,51,61]. To address the challenge of limited resources, research collaborations and partnerships were key to supporting research and RCS initiatives in resource-constrained settings [61]. This allows the sharing of research resources that are available in partner institutions [44].

### Political and economic environment

Four papers (12.5%) highlighted how the wider political and economic environment can both influence RCS and also promote or inhibit research capacity [41,54,60,62]. The researchers highlight how unstable political, economic, and historical contexts affected research partnerships, government buy-in and support for research and RCS, recruitment and retention of university faculty/ researchers and implementation of collaborative research. For instance, for research partnerships/ collaborations to foster real systemic change in RCS, the environment should be secure and stable enough to allow researchers' mobility and free interaction [60]. These macro-level issues vary across contexts in that the agency to affect change often resides in more resourced and stable environments [64].

### Equitable strategic research partnerships

Four papers (12.5%) reported how equitable and strategic research partnerships were central to the RCS efforts [44–46,49]. Partnerships are characterised by power dynamics and control, given the differing expectations and the desire for financial control among partners [49]. Power dynamics between Global North and African partners are more evident, especially when the grants are managed by the Global North partners as this determines the level of autonomy or influence by African institutions in relation to when and how funds are spent [49], which triggers ownership and buy-in, or lack thereof, by the institutional leaders. To attain an equilibrium in research partnerships, there must be fairness in terms of recognition of expertise and research roles of less visible partners [44], and the grants should be flexible to allow investment in local priorities and needs [46].

### Context mechanism and outcome: The causal link

Based on the contexts, mechanism and outcomes that emerged from the review process, the following possible generative causal explanations could be derived. Given the complexity of RCS initiatives which are implemented in complex environments, targeting multiple actors at different levels of individual and institutional capacities, and characterised by varied activities; it is likely that multiple contextual elements will provide a conducive environment where multiple mechanisms will simultaneously be triggered to generate multiple research capacity outcomes as described below.

#### Initial Programme Theory 1

In a context where the university faculty staff have no protected time for research and have limited or no access to well-established physical research infrastructure and equipment and no established research career pathways, it is likely that research capacity strengthening efforts aimed at improving the infrastructure and equipment and research environment including research career pathways and protected time will likely motivate, incentivise, and empower the faculty staff and enact ownership and buy-in among university leaders. This will likely result in the adoption of policies on protected time and research career pathways and improved physical research infrastructure and equipment. In this case, where basic research capacity does not exist, the goal of RCS initiatives will be to build relevant infrastructure and develop relevant research policies. We used the CMO elements to hypothesise that:

**IF** faculty staff have no protected time for research and are based in a university where there is a lack or limited access to well-established physical research infrastructure and equipment and no established research career pathways (C), Research grants and partnerships that support for the upgrade of infrastructure and equipment, and adoption of institutional research policies such as research career pathways and protected time (I), **THEN** policies on protected time and research career pathways will be adopted, and the physical research infrastructure and equipment improved (O), **BECAUSE** this will likely motivate, incentivise, and empower faculty staff and enact ownership and buy-in among university leaders (M).

#### Initial Programme Theory 2

In a context where the university faculty staff have protected time for research and have access to well-established physical research infrastructure and equipment and based in a university with established research career pathways, it is likely that research capacity strengthening efforts aimed at improving the infrastructure and equipment and research environment including research career pathways and protected time will likely motivate, incentivise, and empower the faculty staff. This will likely result in research career and portfolio growth. However, as the university's research portfolio expands, it is possible that the available teaching and administrative resources may be overstretched. As such, RCS efforts would be yielding an unintended negative outcome. We, therefore, hypothesised that:

**IF** faculty staff have protected time for research and are based in a university where there is access to well established physical research infrastructure and equipment and established research career pathways (C) and there are research partnerships between highly capacitated and low capacitated universities/ research institutions that allow sharing of resources such as financial, technical expertise and infrastructure (I), **THEN** the faculty staff will likely record growth in their research career and portfolio, but this may strain their departmental teaching and administrative resources (O), **BECAUSE** the staff will be motivated, incentivised, and empowered to carry out research (M).

#### Initial Programme Theory 3

In a context where the faculty staff have protected time for research, access to well-established physical research infrastructure and equipment and are based in a university with established research policies, systems, and culture, research capacity strengthening efforts will likely motivate, incentivise, and empower faculty staff and spark commitment and enthusiasm among them to conduct high quality and impactful research. This will likely see the faculty staff record career growth in their research and research portfolio, including securing additional research funding and impactful research that improves people's health outcomes and influences policymaking being generated. In this case, a certain level of institutional research capacity is required to strengthen that capacity further, which adds a layer in the complexity frame. Based on the authors' [MNM and JP] experience supporting RCS programmes in Africa, most funding partners supporting RCS initiatives invariably require institutions to meet or demonstrate a minimum capacity level, for instance, in research

governance and grants management, before they can be funded. In such cases, therefore, RCS initiatives will be aimed at further strengthening the existing research capacity. We used the CMO elements to hypothesise that:

**IF** faculty staff have protected time for research, access to well-established physical research infrastructure and equipment and are based in an institution with established institutional research policies, systems, and culture (C) and there are research partnerships between highly capacitated and low capacitated universities/ research institutions that allow sharing of resources such as financial, technical expertise and infrastructure and provide mentorship for emerging researchers (I), **THEN** there will be increased retention of faculty staff and research portfolio, including research funding and generation of impactful research that promotes positive health outcomes and policymaking (O), **BECAUSE** motivation, incentivisation, empowerment, and a sense of commitment and enthusiasm will be enacted among junior and senior faculty staff (M).

### Initial Programme Theory 4

In a context where a university has established equitable partnerships, has access to sustainable long-term funding and is based in a stable political and economic environment, research capacity strengthening efforts may trigger ownership and buy-in by institutional leaders and also trust and respect among researchers because the contextual conditions allow the researchers to address their local research capacity priorities and needs. This will likely result in strengthened research partnerships, research management, and governance systems and the generation of impactful research that improves people's health outcomes and influences policymaking. For these long-term RCS outcomes to be realised, sustained funding, equitable and strategic partnerships, and stable research environments are key. We, therefore, hypothesised that:

**IF** a university has established equitable strategic research partnerships, has access to sustainable long-term funding and is based in a stable political and economic environment (C) and there are research partnerships between highly capacitated and low capacitated universities/ research institutions that allow sharing of resources such as financial, technical expertise and infrastructure (I), **THEN** research partnerships and research management and governance systems will be strengthened, and impactful research that promotes positive health outcomes and policymaking will be generated (O), **BECAUSE** there will be ownership and buy-in among university leaders and trust and respect among staff (M).

### Findings from the Partners' Interviews

Below, we present the three refined initial theories identified from the partners' interview data. For each refined IPT, an 'IF…THEN…BECAUSE' statement has been constructed. Importantly, chunks/ fragments of data (quotes) denoting intervention (I), context (C), mechanism (M) and/ or outcomes (O) were included to demonstrate how the participants' voices were central to the process.

### Programme Theory 1: Motivate, incentivise, empower, and challenge/inspire faculty staff

This subsection refines the initial programme theory 1 derived from the literature review.

Participants shared that in a context where research capacity was non-existent, the goal of the RCS programme was to build it. For instance, if the university did not have the requisite research policies in place, the programme would advocate (I) for the adoption/ adaptation of key research policies such as protected time, research career pathways, gender equality, whistleblower, research ethics and integrity, anti-harassment and anti-bullying. Similarly, in a context where the university did not have well-established physical research infrastructure and equipment, the programme would provide research grants, partnerships and collaborations (I) that promote research capacity sharing and the upgrade of the infrastructure and purchase of critical research equipment. The participants reported that research grants, partnerships and collaborations (I) were a key resource that will motivate, incentivise and empower (M) the faculty staff and leaders towards achieving improved physical research infrastructure and equipment.

…through the research partnerships and those institutional collaborations developed by the DELTAS consortia institutions (I) was to support the establishment of infrastructure and of course purchase of critical research equipment when such did not exist (C). …I know some of the institutions were able to upgrade their infrastructure and equipment. For instance, the university of [name] in Nigeria was able to put up a PhD training centre which now accommodates trainees beyond DELTAS programme (O)…in terms of the research policies, there was no focus on one or two policies, no, the idea was to ensure that the wide range of policies were adopted by the institutions and obviously protected time, research career progression, research integrity and anti-harassment policies were some of the policies that institutions reported adopting (O). [Participant 3]

…in Phase One [of DELTAS programme], research partnerships and collaborations were of course seen as a way of sharing research resources and capacity among partner institutions (I), the only difference is that the partnerships were not intentionally operationalised using the hub and spoke model as in the Phase two of the DELTAS programme. …we included advocacy efforts targeting partner institutions and particularly the leaders (I) to ensure policies around gender equality, research misconduct and integrity, addressing bullying and harassment practices are adopted and implemented (O). [Participant 1]

To see any reasonable change in terms of adoption of research policies and upgrading of research infrastructure and equipment, the participants argued that ownership and buy-in from the university leaders (C) including the Vice Chancellors, Deputy VCs and heads of research departments was needed to ensure that the process was internally led and had the required support from the leaders. The participants shared with institutional buy-in and ownership, the faculty staff would likely be inspired and challenged (M) through the research partnerships and collaborations (by their partners/ collaborators) to adapt/adopt relevant research policies.

…you'll need buy-in and ownership by the leadership for the RCS activity to effectively be delivered and yield positive change. It is not an end in itself but a means through which the desired capacity is achieved. If there's buy-in and ownership by leaders, then you can be assured that you will receive the institutional support you need to effectively implement the DELTAS research capacity-building activities and any policy and systemic changes that might result from that activity. …I think the active learning, knowledge exchange and capacity sharing that happens in the context of research partnerships and collaboration obviously allows institutions to be inspired (M) to improve their research policies. … [Participant 4]

For instance, the [university] was challenged and inspired (M) by the University of [name] to adopt their research integrity policy template and even supported them to customise it. …the idea was simply to ensure such policies are adopted by universities that don't have them already (C). [Participant 1]

The CMO elements were used to formulate the revised initial theory following the 'If…then…because…' format as shown in Box 1 below.

### Box 1. Refined rough programme theory 1

IF a university does not have established research-related policies including protected time, research career pathways, gender equality, whistleblower, research ethics and integrity, anti-harassment and anti-bullying (C1), has no well-established physical research infrastructure and equipment (C2) but there is ownership and buy-in among university leaders (C3), research grants, partnerships and collaborations that support for upgrade of infrastructure and equipment, and advocacy efforts that promote adoption of institutional research policies (I)

THEN relevant research policies will be adopted (O1) and the physical research infrastructure and equipment improved (O2).

BECAUSE the faculty staff will likely be motivated, incentivised, empowered, challenged/ inspired to improve their research practices (M)

The initial draft programme theory was modified as follows: the context was expanded to include gender equality, research ethics and integrity, and anti-harassment and anti-bullying as critical research policies. 'Ownership and buy-in by university leaders' was changed from a mechanism to context. 'Advocacy for the adoption of institutional research policies' was added as a resource. 'Partnership between high and low capacitated institutions' was revised to 'research grants, partnerships, and collaborations aimed at promoting capacity sharing' (resource). 'Faculty staff challenged and inspired to adapt/adopt research policies' was added as a mechanism, and the 'adoption of gender equality, research ethics and integrity, and anti-harassment and anti-bullying' research policies was added as an outcome.

**Programme Theory 2: Motivate, incentivise, and empower faculty staff**

This subsection refines the initial programme theories 2 and 3.

Participants highlighted that in contexts where a university had well-established research infrastructure and equipment (C), and actively implemented and complied with the relevant research policies (e.g., protected time, research career pathways, gender equality, whistleblower, research ethics and integrity, anti-harassment and anti-bullying), the research grants, partnerships and collaborations that support resource sharing (I) among DELTAS partners (I) would both motivate (M1) and incentivise (M2) the faculty staff. The participants explained that in a university context where the research policies are actively implemented and complied, a conducive research environment would be created where it would be possible for the faculty staff to grow in their research career, they would be motivated and incentivised (M) to stay and carry out research in the university (O), grow their research portfolio (O) and generally improve the research culture (O). A participant went on to point out that in a context where a university does not have the requisite research infrastructure and equipment and does not comply with the critical research policies, faculty staff will likely be frustrated by the lack of research infrastructure/ equipment and culture and subsequently leave the institution or research career in entirety.

> <u>Implementation of those research policies and ensuring that the university complies with them is very important</u> (C) if a positive research culture is to be realised. Institutions must ensure that faculty staff are not only protected from negative vices but they are also encouraged and motivated to carry out high-quality research. …<u>making sure that the faculty staff have the research infrastructure they need (C) and then the research environment is enabling for them by having those research policies implemented to the letter</u> (C) can be an <u>incentive in itself</u> (M) to the faculty staff <u>to remain in the university and not look out for opportunities elsewhere</u> (O), but it can also <u>motivate them</u> (M) to conduct <u>cutting-edge research, expand their research work</u> (O) that meets integrity thresholds, and it's <u>at this point that we can claim that the research culture has improved</u> (O) [Participant 2]

> …I think the moment we start talking about increased retention of faculty staff and increased research portfolio, <u>it means that the institution is actively implementing, complying and adhering to those research policies we talked about</u>, <u>that is, the gender equality, the whistleblower, research ethics and integrity, the anti-harassment and anti-bullying policies. That will create a conducive environment where faculty, staff, and researchers are enthusiastic about their research work</u> (C). …the absence of that environment could mean researchers are frustrated and end up exiting the institution or the research career because no one can stand a toxic working environment. …research culture is a huge determinant in any university context, and for researchers to remain enthusiastic about their work, the university must

be committed to addressing those negative practices that might affect the researcher's morale and spirit. I strongly believe that if an institution implements those policies to the letter (C), then we can start seeing a positive culture emerging (O). [Participant 1]

Besides the research grants, partnerships and collaboration used as the vehicle for institutional RCS, participants highlighted that the DELTAS programme also delivered specialised training to university faculty staff and researchers. This included areas such as grant writing, research leadership, networking, research communication, community and policy engagement, and collaboration development and publication. By targeting university faculty staff and researchers with specialised training and mentorship, the assumption was that the faculty staff would in turn use their acquired skills to develop and submit competitive research grants (which would benefit the institution through overheads fees), disseminate research outputs through publications (thus improving the institutional research profile) and expand their networks and collaborations which would – by extension – benefit their institution. Therefore, in a context where the university has established research infrastructure and equipment, and actively implemented and complied with the research policies, the specialised training and mentorship for emerging researchers (I) will likely empower them (M) with the skills to engage in high quality research work, grants writing, research communication and policy engagement resulting in increased research portfolio (O2) and generation of impactful research that promotes positive health outcomes and policymaking (O4). A participant shared the following.

…under the CARTA programme, for instance, the specific target on junior faculty staff and researchers with specialised training and mentorship (I) is to equip them with the skills they need to be able to write and submit competitive research grant applications, effectively lead their research initiatives, expand their research networks and widely disseminate their research outputs [empowerment] (M). …when we empower the faculty staff (M) with grants writing skills, policy engagement, science communication, publication, among others, then we can start seeing the staff carrying out better research and writing winning grant proposals that secure funding and expand their research activities (O) and importantly their cutting edge research having a real impact because policymakers and practitioners will be able to utilise the research evidence (O). …institutions do benefit from overheads since most research grants contribute like 10 or 15 per cent to overheads. Regarding dissemination of research, institutions do benefit whenever their staff publish their work because you obviously elevate the research profile of that institution. [Participant 1]

The assumption that the *faculty staff's research career and portfolio growth* would inadvertently *strain the departmental teaching and administrative resources* was challenged by participants. The participants argued that both research and teaching inform each other, and as the research portfolio expands, the faculty staff are able to actively engage their students and expose them to new knowledge generated through research. The participants also highlighted that all the DELTAS-supported researchers and faculty staff are usually capacity built in grant writing and research leadership. Research personnel costs are therefore factored into those research grant proposals, thus ensuring that once the funding is secured, adequate personnel is in place to implement the research project. A participant shared the following.

…that's a sound assumption, but I beg to differ with the last part where you said that growth in research career and portfolio would inadvertently strain the teaching or administrative resources. The reason why protected time policies exist is to ensure that there is an adequate allocation of time for research. In reality, when the research portfolio is growing, we have observed improvement in the teaching function because the faculty staff are able to bring their research experience to bear in the classroom experience, which is best practice. But the important part is that as the researcher, when you are preparing your research grant application, the finance and grants office will usually factor in all the resources you need to effectively implement your research project, including the personnel. [Participant 2]

The authors' [MNM and JP] experience is that large-scale RCS initiatives (such as DELTAS Africa) usually provide the budget base necessary for recruiting additional staff to support the expanding research portfolio. Small-scale RCS initiatives, however, are potentially more disruptive to teaching because the funding is not adequate enough to both hire additional human resources and implement RCS activities. Based on this, the '*strain the departmental teaching and administrative resources'* outcome was excluded. The CMO elements were used to formulate the revised initial theory following the 'If…then…because…' format as shown in Box 2 below.

### Box 2. Refined rough programme theory 2

IF a university is compliant to research-related policies (e.g., protected time, research career pathways, gender equality, whistleblower, research ethics and integrity, anti-harassment and anti-bullying) (C1), has well-established physical research infrastructure and equipment (C2), research grants, partnerships and collaborations that support resource sharing, and specialised training and mentorship for emerging researchers (I)

THEN in increased retention of faculty staff (O1), increased research portfolio including research funding (O2), improved research culture (O3) and generation of impactful research that promotes positive health outcomes and policymaking (O4)

BECAUSE the faculty staff will likely be motivated, incentivised, and empowered (M).

The initial draft programme theory was modified as follows: the context was expanded to include 'compliance with relevant research policies (e.g., protected time, gender equality, research ethics and integrity, and anti-harassment and anti-bullying); the outcome was expanded to include 'improved research culture,' and the 'strain on departmental teaching and administrative resources' was excluded as an unintended negative outcome. The draft IPT2 and 3 were consequently merged because the contexts and mechanism were similar.

#### Programme Theory 3: Trust, respect, and peace of mind among partners/ faculty staff

This subsection refines the initial programme theory 4.

Participants shared that in a context where a university has established equitable strategic research partnerships (C) and there was a guarantee for long-term funding (C), research grants, partnerships and collaborations that promoted sharing of resources such as financial, technical expertise and infrastructure (I) would activate trust and respect (M) among the partners and the researchers. This would make it possible for the research partnership itself and the research management and governance systems to be strengthened (O). Participants argued that equitable partnerships are important in addressing the actual capacity needs of partners and importantly leveraging each other's strengths. Notably, the participants highlighted that a stable political and economic environment (C) goes hand in hand with the guarantee for long-term research funding since allocation of research funding by the local and national governments is contingent upon the economic performance and the political will and stability. According to the participants, when there is economic and political stability and sustainable funding, the faculty staff are able to enjoy peace of mind (M) knowing that their research activities will not be disrupted and it is this consistent, long-term research effort that generate impactful research that can improve people's health outcomes (O). Overreliance on external philanthropic funding makes the research environment unpredictable, and universities are not able to think long-term in terms of research activities, and this can adversely affect their ability to generate impactful research. This is captured in the quotes below.

…in politically unstable countries, like DRC, you can clearly see that they cannot make strides in any sector leave alone health research. I think having a stable economic and political environment (C) is very important for your peace of mind as a researcher (M). It will mean, then, that you can focus on your research activities undisrupted which is something we often take for granted. To build research capacity which is a long-term exercise needs a politically and economically stable macro environment. …unfortunately, African universities and research institutions overly rely on external funding simply because the local and national governments do not have the resources, and this renders the research environment very unpredictable and unstable (C). …most of the external funding has short cycles of like three to five years with the exception of DELTAS which is running for ten years but even then, some of the consortia that were funded in Phase 1 are not funded in Phase 2. So, it is very difficult to generate anything meaningful with those short funding cycles (C). [Participant 2]

The concept of equitable partnership has a power dynamic and mostly the established universities or institutions that have stable funding sources have the guts to question how equitable or not the partnerships are. …We are talking about engaging as an equal partner, being respected and trusted by partners (M). …because of that respect and trust (M) among the partners, it is possible that the research partnership itself will be strengthened further, and the partner's research management governance processes will also be strengthened in that process (O). …obviously there are practical needs like renumeration, funds for procuring research equipment and all the costs related to research that requires adequate funding. So, if the research environment has stable funding and there is political conflict (C), you can go about your day without worrying (M). As a researcher, you don't want to constantly be worrying about whether you'll get your salary paid or if you will not be able to go to the lab because there's political unrest. …I know of two promising scientists who left Burkina Faso for France because of political instability, and I can say that similar cases are replicated across African countries. [Participant 4]

The CMO elements were used to formulate the revised initial theory following the 'If…then…because…' format as shown in Box 3 below.

### Box 3. Refined rough programme theory 3

IF a university has established equitable strategic research partnerships (C1), has access to sustainable long-term funding (C2) and is based in a stable political and economic environment (C3), research grants, partnerships and collaborations that allow sharing of resources such as financial, technical expertise and infrastructure (I)

THEN the research partnerships, management and governance systems will be strengthened (O1) and impactful research that promotes positive health outcomes and policymaking generated (O2).

BECAUSE trust and respect, and peace of mind will be triggered among partners and faculty staff (M).

The initial draft programme theory was modified as follows: 'peace of mind for researchers' was added as a mechanism (which is triggered in stable political and economic environments), and 'partnership between high and low-capacity institutions' was changed to 'research grants, partnerships, and collaborations aimed at promoting capacity sharing' (resource).

## Discussion

This theory gleaning paper elicited three IPTs representing a diverse range of concepts crucial to the implementation and effectiveness of HRCS programmes implemented in African universities. The IPTs describe HRCS of African universities at different stages of research capacity. For instance, IPT1 describes a university context where research capacity is

almost non-existent and no relevant research policies and procedures are in place. In such a context, the goal of HRCS interventions would be to build research capacity from scratch, for instance, adoption of research policies, development of research infrastructure and acquisition of research equipment. IPT2 and IPT3 describe university contexts where basic research capacity exists. In such contexts, the goal of HRCS interventions would be to strengthen the existing institutional research systems to achieve more long-term research capacity outcomes such as retention of faculty/researchers, growth in research career and portfolio, increased research funding, improved research culture, and generation of impactful health research that has positive effects on people's health and policymaking. This distinction between IPT1 (capacity building) and IPT2/3 (capacity strengthening) is consistent with previous attempts as reported by Dean et al. [69] to distinguish between the terms capacity building and capacity strengthening.

Since many HRCS initiatives implemented in LMICs (including the DELTAS Africa programme) are delivered based on research grants, partnerships and collaborations [70], the IPTs in this synthesis were therefore framed using 'research grants, partnerships and collaborations' as the programme strategy. The research grants, partnerships and collaborations foster the sharing of research funding, expertise, equipment and facilities and best practices among faculty/researchers and their institutions [70]. Some of the direct investments go towards upgrading research infrastructure and purchasing research equipment and reagents [10,39], while the majority of the HRCS strategies focus on individuals (e.g., university leaders, researchers/ faculty) working in the universities. For instance, the HRCS initiative may target university leaders through advocacy efforts to ensure they are supportive of the RCS interventions and buy-in to any research policy changes that come with such interventions [43], the faculty staff/ researchers by empowering them with technical and soft skills to become agents of change within their institutions [40,45], or research management, finance, and grants management support staff whose improved capacity could consequently improve the research, finance, and grants management practices [67,71].

The evidence synthesised in this article suggests that, for HRCS interventions to be more effective in strengthening research capacity in African universities, contextual challenges (e.g., lack of funding for health research, ineffective research policies or lack thereof, lack of institutional support for research and poor research culture, etc.) need to be addressed [16,51] and the university staff empowered, incentivised and motivated [50,53]. A systems approach to HRCS should be used to deliver sustainable impact [72], and by doing so, provide an enabling and conducive research environment where researchers, research and research users can thrive. For instance, the implementation and adherence to research policies (e.g., protected time, research career pathways, among others), and fostering a positive research culture [16,59] guarantees a university environment that is conducive to research. Adherence to research policies will ensure that faculty staff/ researchers are protected against negative vices (e.g., bullying, harassment, gender bias) by ensuring that appropriate channels for reporting and addressing such vices are in place [73]. Importantly, adherence to research policies such as research ethics and integrity would promote good research practices which are critical for promoting people-centred research that has a real impact to policymaking and population health outcomes [74].

The IPTs depict that multiple contexts are necessary for one or more mechanism to be triggered, thus generating one or more research capacity outcomes. For instance, effective implementation and compliance to a career pathway/promotion policy can motivate faculty to grow in their research career, but a friendly research culture that is free from bullying and harassment can motivate them to stay. Staff retention is a key issue that affects long-term HRCS efforts and which require improvements in research capacity and opportunities at systems level and follow-on funding [75]. It is challenging to achieve staff retention in LMICs given the environment characterised by a limited funding base, which affects almost every other aspect [42,75]. Notably, an equitable research partnership will enact trust and respect among researchers/ faculty staff, thus strengthening the partnership further [49] and allowing impactful research to be generated since the partnership will be addressing local priorities and respecting local knowledge [44,46,49]. The primary question is whether equitable partnerships can be achieved by African universities when there is so much reliance on external funding from Global North entities [16,44,56,67]. To achieve the long-term research capacity outcomes, the university staff/ researchers

will also need a stable political and economic environment with adequate research funding, which will both incentivise them and provide peace of mind. For instance, adequate funding may guarantee them decent remuneration and facilitate the procurement of critical research equipment and reagents, and the stable political environment will mean that the research activities will go undisrupted. Undoubtedly, African governments need to invest and sustain long-term funding towards health research and providing a stable political and economic environment where HRCS efforts can yield a sustainable impact [76].

The evidence identifies several HRCS outcomes, which include improved research infrastructure, increased research funding, research career and portfolio growth, staff retention, improved research policies and culture, improved research management and governance, strengthened research partnerships and generation of impactful research [10,16,40–42,56,62,64,65]. Although HRCS initiatives are usually designed to deliver desired positive capacity outcomes, given the complexity of those initiatives and the environment within which they are implemented, unintended negative outcomes might be generated in ways that may not be anticipated. For instance, one of the papers reviewed in this synthesis highlighted that the expansion of the research portfolio in a university could inadvertently affect the teaching function by taking away (stretching) the available teaching resources [55], which is an unintended negative outcome. However, the DELTAS Africa programme designers argued that as the research portfolio expands, ideally, the teaching function and activities get better because the research evidence is brought to bear with the classroom teaching activities. This is practicable, from their perspective, because the DELTAS Africa programme is a large-scale initiative that provides the budget base necessary for recruiting additional staff to support the expanding research portfolio. However, our experience is that small-scale HRCS initiatives are potentially more disruptive to teaching because the funding is inadequate to both hire additional human resources and implement RCS activities.

All the mechanism identified through the realist review phase and the DELTAS programme partners' interviews are consistent with Pawson and Tilley's [20] argument that programmes do not work by themselves; rather, it is the decisions, reasoning, actions and reactions of the individuals directly or indirectly affected by a programme that make it work or not. Even when it comes to upgrading of physical research infrastructure and equipment, research capacity outcomes will only be realised if the reasoning, perception, action or choices of the targeted faculty staff are altered. In other words, the HRCS initiative should provide opportunities, resources or constraints that activate the mechanism responsible for positive research capacity outcomes. The understanding of the generative causal mechanism in HRCS programmes can inform better programme design and implementation and evidence use by policymakers and practitioners [35]. The initial programme theories, thus help them to start thinking about causality.

This synthesis was not free of limitations. The major challenge emanated from the fact that none of the papers included in the literature review had presented findings/arguments using the CMO framing. Our first and last authors' [MNM and JP] professional hunches and experiences in HRCS, particularly in the African context, were helpful in making the linkages between the C, M and Os. The second author's [AH] realist expertise helped to ensure that the articulation of the IPTs was in line with the realist philosophy. Since the synthesis process was iterative, discussions among the co-authors helped to tweak the CMOs multiple times until consensus was reached. Importantly, it is clear, given the long chain of CMOs, that multiple contexts are required for one or more mechanism to be triggered to generate one or more outcomes. As demonstrated in this article, the inability to link a single outcome to a single mechanism and context explains the multifaceted and complex nature of the HRCS initiatives. The high level of abstraction of the theories applied helped to generate few overarching but transferable programme theories, and consequently this helped to avoid generation of overabundance of IPTs as guided by Shearn et al [77]. For instance, instead of focusing on the mechanism triggered by the 'implementation and adherence to anti-bullying and anti-harassing policies' in particular, the focus remained on the 'research policies that foster a conducive research environment/ culture.' The former would have looked at each policy provision(s) as a context and demand eliciting a mechanism for each policy provision(s); this would have probably generated an unmanageable number of IPTs.

In addition to not making a linkage between the CMO elements, some of the papers did not describe all the three C, M and O elements. As shown in the Realist Review section, most of the papers described contexts and outcomes, with a few describing mechanisms. Although this was unsurprising because the papers were not written to bring out CMOs, all the fragmented CMO elements identified across the papers were useful in the theory-gleaning process. The authors also understood that the theory-gleaning process did not need to generate perfect CMO hypotheses but reasonable suppositions about how and why RCS interventions might work since they would be tested and refined in a subsequent primary realist evaluation study – in line with the argument Fick and Muhajarine [78].

This is, to our knowledge, the first fully fledged realist synthesis that has explored how HRCS interventions work to strengthen the institutional research capacities of African universities. The only other published study is a 'light-touch' realist evaluation by Marjanovic et al [79]. These transferrable and overarching IPTs can be applied across different university settings in Africa and potentially other LMIC regions to inform the design, implementation, and evaluation of HRCS interventions, thus ensuring that they can drive sustainable impact. Besides advancing the institutional HRCS discourse and related CMOs, this article also makes a methodological contribution. There are few articles that provide practical guidance on how to develop robust IPTs, and this paper addresses that gap specifically in the HRCS context.

## Conclusion

Institutional HRCS is undoubtedly a complex and multifaceted activity that involves intervening at different levels (individual, institutional and systems) to generate research capacity outcomes which are moderated by multiple contextual aspects. The value of complexity-aware approaches like realist evaluation in understanding how HRCS interventions work in diverse African university settings cannot be overstated. This article sheds light on the critical mechanisms, contexts, and outcomes that could drive health research capacity at African universities. The insights presented in this article can guide future HRCS evaluations and help to establish a clearer understanding of the generative mechanisms underpinning effective institutional-level research capacity strengthening interventions. To drive sustainable improvements in health research capacity at African universities, there is a need to intervene at individual, institutional and policy levels with the aim of generating systemic changes.

The IPTs presented in this article will be tested through a primary realist evaluation covering the DELTAS Africa Programme (Phase I) participants. Based on the emerging evidence, the IPTs will be confirmed, refuted or refined, and new programme theories potentially identified. The realist evaluation will, therefore, expand on the CMO concepts described in this article and provide insights into the application of realist evaluations in HRCS settings.

## Supporting information

**S1 File. Quality and relevance appraisal form.**
(DOCX)

**S2 File. Full list of screened papers.**
(XLSX)

**S3 File. List of included papers.**
(XLSX)

**S4 File. The PRISMA checklist.**
(DOCX)

## Author contributions

**Conceptualization:** Meshack Nzesei Mutua, Andrew Harding, Justin Pulford.

**Data curation:** Meshack Nzesei Mutua.

**Formal analysis:** Meshack Nzesei Mutua, Ferdinand C. Mukumbang.

**Funding acquisition:** Justin Pulford.

**Investigation:** Meshack Nzesei Mutua, Andrew Harding.

**Methodology:** Meshack Nzesei Mutua, Andrew Harding, Ferdinand C. Mukumbang, Justin Pulford.

**Project administration:** Meshack Nzesei Mutua.

**Supervision:** Andrew Harding, Ferdinand C. Mukumbang, Justin Pulford.

**Visualization:** Ferdinand C. Mukumbang.

**Writing – original draft:** Meshack Nzesei Mutua.

**Writing – review & editing:** Andrew Harding, Ferdinand C. Mukumbang, Justin Pulford.

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
