## [Decision Letter · Decision Letter 0]

4 Mar 2025

Dear Dr.  Mutua ,

Thank you for submitting your manuscript to PLOS ONE. After careful consideration, we feel that it has merit but does not fully meet PLOS ONE’s publication criteria as it currently stands. Therefore, we invite you to submit a revised version of the manuscript that addresses the points raised during the review process.

We look forward to receiving your revised manuscript.

Kind regards,

Himanshu Sekhar Rout, PhD

Academic Editor

PLOS ONE

Journal Requirements:

“Medical Research Council (MRC) Doctoral Training Award”

6. We note you have included a table to which you do not refer in the text of your manuscript. Please ensure that you refer to Table 1 and 2 in your text; if accepted, production will need this reference to link the reader to the Table.

7. Please include captions for your Supporting Information files at the end of your manuscript, and update any in-text citations to match accordingly. Please see our Supporting Information guidelines for more information: http://journals.plos.org/plosone/s/supporting-information .

8. As required by our policy on Data Availability, please ensure your manuscript or supplementary information includes the following:

Additional Editor Comments:

**Major Revision**

Reviewers' comments:

Reviewer's Responses to Questions

**Comments to the Author**

1. Is the manuscript technically sound, and do the data support the conclusions?

Reviewer #1: Yes

Reviewer #2: Yes

2. Has the statistical analysis been performed appropriately and rigorously?

Reviewer #1: N/A

Reviewer #2: N/A

3. Have the authors made all data underlying the findings in their manuscript fully available?

Reviewer #1: Yes

Reviewer #2: Yes

4. Is the manuscript presented in an intelligible fashion and written in standard English?

Reviewer #1: Yes

Reviewer #2: Yes

Reviewer #1: Thank you for the opportunity to review this paper.

This is an important topic & this paper is a good one in its field. I have made some comments on the manuscript that I hope can strengthen the quality of the paper:

1- Overall, I found the paper difficult to follow. I suggest the authors scale back the questions and analysis completed in this paper to provide more space to introduce the concepts. Hence, findings should be modified and structured based on the questions of the research.

2- Keywords should be modified base on MeSH.

3- As it mentioned in page 8 line 197 thematic analysis was applied in this research but it is not any table in the body of article to show main Themes and Subthemes of finding of this review.

4- limitations of the study is not mentioned in the paper.

Reviewer #2: The paper is an important contribution to the research capacity strengthening efforts. It has effectively used realist evaluation framework. However, the paper needs to be revised and made concise significantly. The paper is difficult to read in its current format. Many redundant information needs to be removed. The findings are presented in present tense and therefore, reads like a discussion point. The sentence structure needs to be changed in several places throughout the paper; I have not highlighted every place which needs this change. Some of the domains parsed out in the findings can also be lumped together. This can perhaps help to eliminate redundancy. There are other specific comments in various parts of the paper that have been uploaded in the pdf.

**Do you want your identity to be public for this peer review?** For information about this choice, including consent withdrawal, please see our Privacy Policy

Reviewer #1: No

Reviewer #2: No

---

## [Author Response · Author response to Decision Letter 1]

29 Apr 2025

Please check the 'Response to Reviewers' document for the detailed description of how each of the editor's and reviewers' comments has been addressed.

---

## [Decision Letter · Decision Letter 1]

31 Jul 2025

Eliciting initial programme theories for a health research capacity strengthening initiative targeting African universities: A realist synthesis

PONE-D-24-49110R1

Dear Dr. Mutua,

We’re pleased to inform you that your manuscript has been judged scientifically suitable for publication and will be formally accepted for publication once it meets all outstanding technical requirements.

Kind regards,

Himanshu Sekhar Rout, PhD

Academic Editor

PLOS ONE

Additional Editor Comments (optional):

Reviewers' comments:

Reviewer's Responses to Questions

**Comments to the Author**

Reviewer #1: (No Response)

Reviewer #3: All comments have been addressed

2. Is the manuscript technically sound, and do the data support the conclusions?

Reviewer #1: Partly

Reviewer #3: Yes

3. Has the statistical analysis been performed appropriately and rigorously?

Reviewer #1: N/A

Reviewer #3: Yes

4. Have the authors made all data underlying the findings in their manuscript fully available?

Reviewer #1: Yes

Reviewer #3: Yes

5. Is the manuscript presented in an intelligible fashion and written in standard English?

Reviewer #1: Yes

Reviewer #3: Yes

Reviewer #1: (No Response)

Reviewer #3: (No Response)

**Do you want your identity to be public for this peer review?** For information about this choice, including consent withdrawal, please see our Privacy Policy

Reviewer #1: No

Reviewer #3: No

---

## [Editor Report · Acceptance letter]

PONE-D-24-49110R1

PLOS ONE

Dear Dr. Mutua,

I'm pleased to inform you that your manuscript has been deemed suitable for publication in PLOS ONE. Congratulations! Your manuscript is now being handed over to our production team.

Kind regards,

on behalf of

Professor Himanshu Sekhar Rout

Academic Editor

PLOS ONE